# Adequacy of Nutrient Intake and Malnutrition Risk in Older Adults: Findings from the Diet and Healthy Aging Cohort Study

**DOI:** 10.3390/nu15153446

**Published:** 2023-08-04

**Authors:** Kaisy Xinhong Ye, Lina Sun, Su Lin Lim, Jialiang Li, Brian K. Kennedy, Andrea Britta Maier, Lei Feng

**Affiliations:** 1Department of Psychological Medicine, Yong Loo Lin School of Medicine, National University of Singapore, Singapore 119007, Singapore; kaisy.ye@nus.edu.sg; 2Centre for Healthy Longevity, @AgeSingapore, National University Health System, Singapore 119228, Singapore; bkennedy@nus.edu.sg (B.K.K.); a.maier@nus.edu.sg (A.B.M.); 3School of Anesthesiology, Weifang Medical University, Weifang 261053, China; sln@wfmc.edu.cn; 4Department of Dietetics, National University Hospital, Singapore 118177, Singapore; su_lin_lim@nuhs.edu.sg; 5Department of Statistics & Applied Probability, National University of Singapore, Singapore 119077, Singapore; jialiang@nus.edu.sg; 6Healthy Longevity Translational Research Program, Yong Loo Lin School of Medicine, National University of Singapore, Singapore 119077, Singapore; 7Department of Biochemistry, Yong Loo Lin School of Medicine, National University of Singapore, Singapore 119077, Singapore; 8Department of Human Movement Sciences, @AgeAmsterdam, Amsterdam Movement Sciences, Vrije Universiteit Amsterdam, 1081 HV Amsterdam, The Netherlands

**Keywords:** nutrient, malnutrition, aged, nutritional assessment, Asian

## Abstract

There is a lack of data on the adequacy of nutrient intake and prevalence of malnutrition risk in Asian populations. The aim was to report on the nutrient intake and prevalence of malnutrition risk in a community sample of older adults in Singapore. Analysis was performed on 738 (*n* = 206 male, *n* = 532 male, aged 67.6 ± 6.0 years) adults 60 years and above. Intakes of macro- and micronutrients were evaluated against the Recommended Dietary Allowances (RDAs). Malnutrition risk was assessed using the Nutrition Screening Initiative Determine Your Nutritional Health checklist. It was found that 90.5% older adults exceeded the sugar intake, 68.5% males and 57.1% females exceeded the intake limit for saturated fat, and 33% males had inadequate dietary fiber intake when compared to the RDAs. Inadequate dietary calcium intake was found in 49.5% males and 55.3% females. There were 22.3% of older adults at moderate to high malnutrition risk. Singaporean older adults need to reduce their dietary intakes of sugar and saturated fat and increase their intakes in dietary fiber and calcium. Current findings provide public health awareness on the importance of healthy eating and will facilitate decision making by health promotors to deliver targeted nutrition care programs.

## 1. Introduction

The older population represents the largest demographic group at risk of malnutrition [1]. It has been reported in the Western community that 10–30% of older adults aged 60 and above have malnutrition [2]. Malnutrition occurs when there is a deficiency, excess or imbalance in dietary intake that prevents normal bodily function, and leads to adverse clinical outcomes [3]. Malnutrition increases risks for morbidity and mortality [4]. Changes in nutritional status among older adults are insidious, often left unrecognized until pronounced physical changes are apparent. Therefore, screening and identifying nutrition problems among community older adults is of utmost importance. Early detection will not only allow public health professionals to address nutrition problems at its earliest stage, but also prevent future incidence of nutrition problems, and mitigate healthcare burden associated with a growing older adult population.

Possible reasons for malnutrition include age-related changes such as alterations in taste, poorer digestion, and chewing and swallowing difficulties, which may reduce ones motivations to eat or drink [5]. Ageism, and reduced mobility may cause older people to become socially isolated and lonely. Social isolation and loneliness are associated with reduced appetite [6], decreased food intakes [6,7], and increased malnutrition risk among older adults [8]. Moreover, poor dietary choices are often seen in socially isolated older adults, who tend to consume lesser fruits and vegetables [9] and have lower dietary variety [10]. Older adults are more likely to experience financial hardship due to retirement, out-living savings [11], and less opportunities for employment [12]. A diet that includes fruit, vegetables and fish is often perceived as a luxury for older adults on constraining food budgets [13]. To lower the cost of living, one might choose less heathy foods that are often high in saturated fat, sugar, and sodium and low in fiber and nutritional value.

Despite concerns over poor nutrition and the risk of malnutrition in the older adult population, there is limited data in this vulnerable population group in Asian countries. To date, no studies have looked at the adequacy of nutrient intakes and malnutrition risk of community-dwelling older adults aged 60 and above in Singapore. As a result, this study aimed to determine nutrient intake and malnutrition risk among community-dwelling older adults in Singapore. Given the difference in nutrition requirements between sex, males and females were investigated separately. Dietary practices and nutrition knowledge were also surveyed to understand the observed dietary intake pattern.

## 2. Materials and Methods

### 2.1. Study Design

Data were generated from the Diet and Healthy Ageing (DaHA) cohort, which is a prospective population-based study established between 2011 and 2017 [14]. The cohort recruited 1060 community-dwelling Singaporean or permanent residents aged 60 years and above. Recruitments were from the Jurong region of Singapore. Interviews were conducted in-person at the Training and Research Academy at Jurong Point (TaRA@JP) by trained research nurses. Structured questionnaires were used to collect information on demographic characteristics and dietary information.

### 2.2. Dietary Intakes

Habitual dietary intakes were measured using a validated semi-quantitative food frequency questionnaire (FFQ) developed by the Singapore Health Promotion Board (HPB) [15]. There are more than 300 food items in the FFQ designed to capture the calories, total fat, types of fats, cholesterol, and other nutrients consumed by the participants. The FFQ was administered by research nurses who were trained by an experienced research dietitian. Subjects were to report number of times the food is consumed per day, week, or month. Portion size pictures/food models and the Compendium of Food Pictures [16] were used to guide responses. Participants were asked for the venue at which food was consumed and cooking methods (e.g., steamed, stir-fried, curry without coconut) to consider for the different nutrient compositions. The intakes of various macro- and micronutrients were calculated based on the information obtained from the FFQ using a database system developed by the HPB. The level of intakes of nutrients were evaluated against the Recommended Dietary Allowances (RDAs) for healthy individuals and the Dietary Guidelines developed by the Health Promotion Board [17,18,19,20]. For vitamins and minerals, intake of less than 70% of the respective RDA is defined as insufficient [21]. For the intakes of polyunsaturated fatty acid (PUFA), monounsaturated fatty acid (MUFA) and saturated fat (SFA), it is recommended that the ratio for all adults is 1.00: 1.00: 1.00 (P: M: S ratio) [19].

### 2.3. Malnutrition Risk Screening

The malnutrition risk screening tool, Nutrition Screening Initiative Determine Your Nutritional Health (NSI Determine) checklist, was used to assess risk for malnutrition among older adults. The tool comprises of 10 checklist items on warning signs for poor nutrition [22]. Based on the aggregate point results, one can be classified as having low nutritional risk (0 to 2 points), moderate nutritional risk (3 to 5 points) or high nutritional risk (6 points and above). The greater the scores, the greater the risk of malnutrition.

### 2.4. Dietary Practices

Dietary practices such as choices of fats and oils (blended vegetable oil, polyunsaturated oil, monounsaturated oil, saturated fat or none) used in cooking, milk (creamer, sweetened condensed, evaporated, full cream, low-fat, skimmed, whitener or none) used in beverages and the habit of asking for less sugar/less sweet when ordering beverages and desserts were also captured to understand health behaviors related to the nutrient intakes.

### 2.5. Nutrition Knowledges

Participants were tested upon their nutrition knowledge on the relationship between fat, salt, dietary fiber, calcium, iron and preserved foods and certain diseases. A total of 12 questions were administered with responses being ‘true’, ‘false’ or ‘don’t know. One point was awarded for each correct response, and no point was awarded for an incorrect response or non-response. The level of nutrition knowledge was defined as either low (0 to 3), moderate (4 to 6) or high (7 or above) based on the scores. Participants were also asked how much attention they pay to dietary and nutrition information and their most important sources of nutrition information (e.g., internet, friends, medical practitioners). The questionnaire was designed by an internal research dietitian and can be found in Appendix A.

### 2.6. Statistical Methods

Only participants that completed the FFQ were included in the analysis. Participants that had inconsistencies in reporting between dietary practice items and FFQ items were excluded from the analysis. All statistical analyses were conducted in Stata (15.1, StataCorp LLC, College Station, TX, USA). Continuous data were tested for normality. Descriptive statistics were used to analyze the demographic and anthropometry data of the study participants. In the cases of skewed data, the median and interquartile range (IQR) were presented instead of the mean and standard deviation (SD). Sex differences in their median nutrient intakes and proportion of DRI consumed were compared using the Mann–Whitney test and Chi-squared test, respectively. The level of significance was set at *p* ≤ 0.05 for all statistical procedures.

## 3. Results

After removal of those that did not complete the interview (n = 44), or had repeated interviews (n = 6) or dietary information missing or incorrect (n = 272), 738 participants were included in this cross-sectional analysis. A flowchart of the selection process can be seen in Appendix A. Participant demographic characteristics are shown in Table 1. The mean age was 67.6 years (SD = 6.0, range: 60–92). Participants were mostly females (72.1%) and Chinese (95.7%). The majority of older adults lived with family. The average number of years of schooling was 6.1 ± 4.3. Around half of the older adults reported to have medically diagnosed high blood pressure (48.4%) and high cholesterol (52.6%). The prevalence of diabetes was found to be 18%. The majority (59.9%) had healthy weights.

### 3.1. Macro- and Micronutrient Intakes

Table 2 shows the intake levels with comparison to the Recommended Dietary Allowances (RDAs). The percentages of males and females meeting the RDA for energy and each of the macro- and micronutrients can be seen in Figure 1. Both males and females had median energy intake (males: 2258 kcal, females: 2058 kcal) within the RDA range based on different physical activity levels. The majority of older adults met the RDA for protein, total fat, and carbohydrate. A proportion of 68.5% males and 57.1% females exceeded the limit for SFAs, PUFAs and MUFAs. The P:M:S ratios for both males and females were 0.50:1.00:1.00 (Table 2). The majority (males: 59.7%, females: 66.7%) had a median intake of cholesterol within the recommended limit of 300 mg daily. Almost all males (87.9%) and females (91.5%) exceeded the intake limit for sugar. The majority of older adults met the requirements for vitamin A (males: 82.5%, females: 88.5%), vitamin C (males: 84%, females: 94%) and iron (males: 99.0%, females: 98.7%). A proportion of 50.5% of males and 44.7% of females had calcium intakes lesser than 70% of their RDA. The majority of females (85.7%) met the RDA for dietary fiber, while 33% of males had inadequate dietary fiber intake. In the sensitivity analysis, one participant with severe cognitive impairment was removed from the analysis to reduce recall bias, but this did not change the results. 

### 3.2. Malnutrition Risk

The NSI Determine found that 15.9% of all, 17.0% of males and 15.4% in females were at moderate nutritional risk. A high nutritional risk was found in 6.4% of participants (males: 5.8%, females: 6.6%). No statistically significant differences in risk were observed on sex (χ2(2) = 0.3783, *p* < 0.828). The questions with the most positive answers included taking three or more different prescribed or over-the-counter drugs a day (30.5%), eating alone most of the time (24.1%), or having an illness or condition that made them change the kind and/or amount of food they eat (22.2%) (Table 3).

### 3.3. Dietary Practices

Table 4 shows the dietary practices among older adults. Monounsaturated oil (e.g., olive oil and canola oil) (40.7%) was the most commonly used oil for cooking at home, followed by polyunsaturated oils (e.g., corn oil and sunflower oil) (36.0%) and blended vegetable oil (22.1%). Only a small number of older adults used saturated oil such as butter, ghee and lard for cooking at home (1.2%). There was no difference in the choices of oil/fat between sex. The most common types of milk/milk substitute used in beverages (tea, coffee, malt beverages) were sweetened condensed milk (19.0%) and creamer (18.8%). Low-fat milk/powder (12.2%) was used more widely than full cream milk/powder (5.0%) and skimmed milk/powder (2.2%). A proportion of 5.2% used evaporated milk, whereas 38.4% of the older adults did not add any milk/milk substitutes to their beverages. Males (21.9%) were more likely to use sweetened condensed milk than females (17.5%), and females (13.2%) were more likely to used low-fat milk/powder than males (9.9%). When ordering beverages or desserts, 43.9% of males and 46.9% of females would ask for less sugar/less sweet options.

### 3.4. Nutrition Knowledge

As shown in Table 5, 57.3% of all participants scored 7 or above out of a total score of 12, with no difference in nutrition knowledge between sex (χ2(2) = 4.4421, *p* = 0.108). A percentage of 37.5% of participants had a moderate, and 5.1% had a low level of nutrition knowledge. A proportion of 47.8% reported that they always pay attention to dietary and nutrition information, followed by 34.5%, sometimes and 14.1%, seldomly pay attention to dietary and nutrition information. No difference was found between sex (χ2(3) = 3.3430, *p* = 0.342) The top three most important sources of nutrition information for the older adults were book/newspaper/magazine, TV/radio, internet (77.4%), relatives, friends and neighbors (32.0%), and medical practitioners, nurses, dietitians (16.5%) (Table 6).

## 4. Discussion

Older adults are more vulnerable to nutritional deficiencies and unhealthy eating. Globally, the prevalence of nutritional deficiency or malnutrition ranges from 0.8% to 24.6% [23]. Malnutrition can lead to poor wound healing, increased propensity for developing infections [5] and anemia [24], while poor dietary choices along with unhealthy lifestyles are associated with an increased risk of diabetes [25], hypertension [26], heart diseases [27] and dementia [28,29]. To promote good nutrition and provide optimal care in community-dwelling older adults, one must first identify nutrition problems, and then implement nutrition strategies to alleviate the effect of poor nutrition and prevent future incidence. Yet, there is still a lack of data on the adequacy of nutrient intake and prevalence of malnutrition risk in older adults in Asian countries. Therefore, the present study combined detailed clinical and dietary measures, and malnutrition risk screening to make inference on the nutritional status of community-dwelling older adults in Singapore.

In the current sampling cohort of older adults, more than 50% of older adults met the RDA for energy, protein, vitamin A, vitamin C and iron. These results are encouraging as previous studies have suggested that Singaporean older adults are likely to be deficient in protein, vitamins and minerals [17,30] However, there were also some less optimistic findings. It was found that almost half of older adults failed to meet calcium recommendations. Calcium has an important role in the prevention of bone mass loss and osteoporotic fractures [31,32]. Calcium absorption is impaired in the later decades of life, and requirements increases from 800 mg to 1000 mg per day after the age of 50. Ho et al. (2004) suggested that at least 900 mg per day of dietary calcium intake is needed to prevent bone loss in Asian postmenopausal women [33]. According to the National Nutrition Survey (NNS) 2010, a low calcium intake could be due to the low intakes of dairy foods such as milk, yoghurt, and cheese in the older adults [21]. Moreover, it was reported that 50% of Singaporean residents aged 18 to 69 do not consume any milk, with older adults consuming the least amount of milk [21]. Moreover, the older adults reported not enjoying drinking milk and lacked awareness of alternative calcium sources or even the existence of calcium [21]. Therefore, there is an urgent need for public health measures to prevent health consequences associated with calcium deficiencies. This could be in the form of health education or the prescription of calcium and vitamin D supplements by health professionals for those at-risk groups.

Furthermore, Singaporean older adults are exceeding the recommended limits for saturated fat (SFA) and sugar and have comparably low intakes for polyunsaturated fat (PUFA) and monounsaturated fat (MUFA). A diet high in SFAs and sugar, and low in PUFAs and MUFAs is associated with higher blood pressure [34] and cholesterol [35], and increased risks for obesity, type 2 diabetes, heart diseases [27], and cognitive decline [36,37]. Around 50% of older adults were found to have high blood pressure and high cholesterol in this study, highlighting the need for public health intervention. Based on the dietary practice survey, we did not identify sources of saturated fat. However, the NNS 2010 suggested that saturated fat intakes were likely to come from plant-based sources such as coconut oil and coconut milk, which are commonly used in local Malay and Indian dishes. Other major contributors of saturated fat included biscuits, pastries, cakes, snacks and local snacks [21]. These foods are also high in added sugar. The dietary guideline for Singaporeans suggests that consumption of sugar should be no more than 10% of the daily energy intake [38], but 90.5% of older adults exceeded this intake limit. This could be due to the common dietary practices found in the older adults, such as adding sweetened condensed milk and sugar to beverages and desserts. The prevalence of type 2 diabetes was found to be 18% in the representative Singaporean older adult population, which was very close to the global estimate of 19.3% in adults 65 years or older [39]. We are one of the rare studies to provide regional sources of diabetes prevalence on those aged 65 years and over, and the evidence continues to point to diabetes as a significant global chronic disease burden in ageing populations.

Dietary fiber has important roles to play in lowering blood cholesterol and sugar, and hence reducing the risk of heart diseases [40,41] and type 2 diabetes [42]. Dietary fiber can be found abundantly in whole-grain products, fruits, vegetables, legumes, and nuts and seeds. It was found that 33% male older adults had inadequate dietary fiber intake and were significantly more likely to have insufficient dietary fiber intake compared to female older adults. According to NNS 2010, three in four adult Singapore residents could not meet the Dietary Guidelines of eating at least two servings of fruit daily, and seven in ten could not meet the two-servings-a-day recommendation for vegetables. Moreover, a recent study found that one in every four Singaporeans would choose to dine out every day [43]. A higher rate of eating out has been associated with poorer diet quality, characterized by greater energy, sugar, sodium and SFA intakes and lower dietary fiber, fruit, vegetable, and micronutrient intakes [44]. Whether the same eating-out pattern is found in the older adult population requires further investigation.

There are many factors affecting the nutritional intake and dietary choices among the older people. For example, living alone or with others would influence dietary habits and quality of life among the older adult population. When cooking for one, older adults may choose quick, nutritionally suboptimal meals such as tea and toast over cooking and enjoying a ‘proper meal’ [45]. The small proportion of older adults living alone in the present study could partly explain the low proportions of micronutrient deficiencies identified. Nutritional knowledge is another factor shown to positively affect eating in older adults [46]. Despite low education levels in our sample, over 90% of them had moderate to high levels of nutritional knowledge, and majority reported that they pay attention to dietary and nutrition information. However, this could not explain for the high saturated fat and sugar intakes and low intakes of calcium and dietary fiber in males in the older adults. It has been suggested that nutritional knowledge could only explain one’s ability to understand food-related and nutrition-related terminology and not the attitudes and beliefs around the individual’s eating behavior toward food. Studies have also suggested that healthy eating in older adults is multi-faceted and is determined by both individual (e.g., age, sex, education, beliefs and behaviors) and collective (accessible food labels, food shopping environment, social support) factors [47]. As a result, studies are warranted to understand contributors to healthy eating to facilitate the development of programs and services designed to encourage healthy eating.

According to the NSI Determine Your Health checklist, 6.4% of the older adults were at high malnutrition risk. This is consistent with previous findings from the Singapore Longitudinal Ageing Study (SLAS) [48,49]. However, there was a reduction in the percentage of moderate risk compared to SLAS (15.9% vs. 24.7%) [49]. Given that both DaHA and SLAS used the NSI checklist for the screening of malnutrition risk, the discrepancies could be due to the different recruitment periods between studies. The recruitment periods for SLAS were 2003 to 2004, and 2009 to 2013, whereas the recruitment period for DaHA was 2011 to 2017. Therefore, the finding from the latter could be reflective of a reduction in malnutrition risk in community-dwelling older adults. Compared to other malnutrition screening tools, the NSI checklist had the advantage of being quick and less invasive (i.e., does not require blood indicators). It also helped to raise awareness in older adults of factors that affect their nutritional health and encourage them to improve their dietary and lifestyle habits to reduce their risk of nutrition-related health problems. The malnutrition screening will also help construct public health interventions, which will incorporate risk factors identified to construct more targeted nutritional health interventions for the local older adult population.

## 5. Conclusions

The current study reported, for the first time, the adequacy of nutrient intakes and malnutrition risk of the community-dwelling older adult population in Singapore. Attention is needed towards the excessive intakes of saturated fat and sugar, and the inadequate intakes of calcium and dietary fiber from diet alone. Nutrition advice for older adults needs to focus on the selection of nutrient-dense foods, and to replace excessive saturated fat and sugar intakes with intakes of dairy and high-fiber products while ensuring nutrient adequacy. Screening for calcium deficiency should be part of routine health check-ups, along with nutrition education on the importance of calcium in a healthy balanced diet. Therefore, the early identification and understanding of the nutrition needs is the key to prevention of malnutrition risk and associated diseases in this growing segment of the population.

## 6. Limitations and Strengths

The current study was not without limitations. First, there was a potential bias in the recruitment procedure. Recruitments were conducted in community settings and those who were most likely to be malnourished were less inclined to attend and be recruited into the study. However, subjects were offered financial incentives to promote participations. Second, FFQs are not the best dietary assessment tool for older adults due to their reliance on memory, and the results may be influenced by factors such as social desirability bias. However, the FFQ used in this study was carried across from the NNS 2010 survey, which has been validated in the Singapore population. Third, intakes for sodium (added salt), which taken in excess is associated with high blood pressure and cardiovascular risk [50], was not examined. Fourth, we did not look at supplementation use, but our findings suggest that nutrient intake from diet alone was adequate for most micronutrients except calcium. Nevertheless, the present study had its strengths. It is one of the very rare studies in Asian countries, and the first in Singapore, to report on the nutrient intake of community older adults. Factors that could have contributed to the observed nutrition intake were also examined and served importance in public health promotion.

## Figures and Tables

**Figure 1 nutrients-15-03446-f001:**
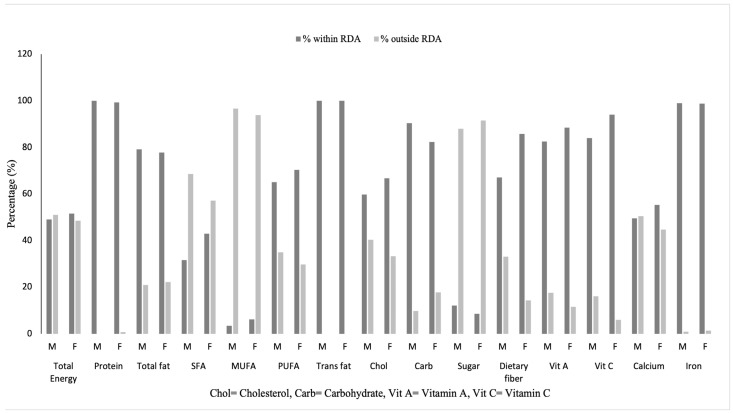
Percentages of males and females meeting the Recommended Dietary Allowances (RDA).

**Table 1 nutrients-15-03446-t001:** Participants (*n* = 738) demographic characteristics.

	All	Male	Female
*n* (%)	738 (100)	206 (27.9)	532 (72.1)
Age, year ± SD	67.6 ± 6.0	68.8 ± 6.2	67.1 ± 5.8
Race, *n* (%)			
Chinese	706 (95.7)	189 (91.8)	517 (97.2)
Malay	6 (0.8)	5 (2.4)	1 (0.2)
Indian	15 (2.0)	7 (3.4)	8 (1.5)
Others	11 (1.5)	5 (2.4)	6 (1.1)
Living status, *n* (%)			
Alone	49 (6.7)	7 (3.4)	42 (7.9)
With family	688 (93.4)	199 (96.6)	489 (92.1)
Schooling, year ± SD	6.1 ± 4.3	7.9 ± 0.31	5.4 ± 0.17
Currently working, *n* (%)	142 (19.4)	49 (24.1)	93 (17.6)
Physical activity, *n* (%)	615 (83.7)	173 (84.4)	442 (83.4)
Current smokers, *n* (%)	40 (5.4)	30 (14.6)	10 (1.9)
Alcohol drinkers, *n* (%)	68 (9.2)	33 (16.0)	35 (6.6)
High blood pressure, *n* (%)	357 (48.4)	102 (49.5)	255 (47.9)
High cholesterol, *n* (%)	388 (52.6)	108 (52.4)	280 (52.6)
Diabetes, *n* (%)	133 (18.0)	41 (19.9)	92 (17.3)
Stroke, *n* (%)	26 (3.5)	12 (5.8)	14 (2.6)
GDS-15, mean ± SD	1.3 ± 2.0	1.36 ± 0.2	1.3 ± 0.1
GAI-20, mean ± SD	1.1 ± 2.5	0.8 ± 0.2	1.2 ± 0.1
MMSE, mean ± SD *	27.7 ± 2.5	28.1 ± 0.1	27.5 ± 0.1
Weight, kg ± SD	59.7 ± 10.6	65.2 ± 10.0	57.6 ± 10.1
Height, m ± SD	1.57 ± 0.078	1.65 ± 0.065	1.54 ± 0.055
BMI, kg/m^2^ ± SD	24.2 ± 3.7	23.9 ± 3.3	24.3 ± 3.9
BMI category (WHO)			
Underweight < 23.0	24 (3.3)	6 (2.9)	18 (3.4)
Normal 23.0–28.0	438 (59.9)	129 (63.2)	309 (58.6)
Overweight > 28.0	269 (36.8)	69 (33.8)	200 (38.0)

Abbreviations: *n*, number; SD, Standard Deviation; HDB: Housing and Development Board; BMI, Body Mass Index; GDS-15, 15-item Geriatric Depression Scale; GAI-20; 20-item Geriatric Anxiety Inventor; WHO: World Health Organization; CDH: United States National Research Council Committee on Diet and Health. * MMSE scores were re-calculated as out of a total score of 30.

**Table 2 nutrients-15-03446-t002:** Macro- and micronutrient intakes with comparison to the Recommended Dietary Allowances.

Nutrients	25%	Median	75%	% Energy Intake	RDA	% within RDA	% under RDA	% over RDA	*p*-Value (Mann–Whitney) ^	Chi-Squared Test #
Total Energy, E (kcal)										
Male	1743	2258	2853	-	1885–3070	49.0	32.0	18.9	0.016	-
Female	1656	2058	2633	-	1570–2560	51.5	20.9	27.6		
Protein (g)					10–15% of total E					
Male	67.1	88.4	113	15.6		36.4	0.0	63.6	0.263	-
Female	63.6	87.0	107	16.1		31.4	0.7	67.9		
Total fat (g)					20–35% of total E					
Male	57.6	74.0	99.9	30.4		79.1	2.9	18.0	0.025	-
Female	51.2	68.9	92.0	29.9		77.8	5.6	16.5		
Saturated fat (g)					<10% of total E					χ2(1) = 7.9260, *p* = 0.005
Male	19.7	27.2	36.7	11.1		31.6	-	68.5	0.002	
Female	17.7	24.3	32.8	10.5		42.9	-	57.1		
Monounsaturated fat (g)					15–20% of total E					
Male	19.9	27.5	37.5	11.3		-	-	-	0.026	-
Female	19.0	25.2	35.0	11.1		-	-	-		
Polyunsaturated fat (g)					5–10% of total E					
Male	10.0	14.0	19.9	5.7		-	-	-	0.511	-
Female	10.2	13.8	18.7	6.0		-	-	-		
P: M: S ratio					1.00: 1.00: 1.00					
Male	0.50:1.00:1.00			-		-	-	-	-	-
Female	0.50:1.00:1.00			-		-	-	-		
Trans fat (g)				negligible intake	0% of total E					
Male	0.246	0.361	0.476			100		0	-	-
Female	0.251	0.338	0.470			100	-	0		
Cholesterol (mg)									0.018	χ2(1) = 3.2076, *p* = 0.073
Male	193	270	372		<300	59.7	-	40.3		
Female	170	250	326		<300	66.7	-	33.3		
Carbohydrate (g)					45–65% of total E					
Male	234	297	389	53.3		90.3	5.8	3.9	0.010	-
Female	222	271	343	53.3		82.3	11.7	6.0		
Sugar (g)					keep at minimal					χ2(1) = 2.3389, *p* = 0.126
Male	61.0	80.9	107	14.7		12.1	-	87.9	0.556	
Female	58.4	77.8	102	15.3		8.5	-	91.5		
Dietary fiber (g)										χ2(1) = 33.1500, *p <* 0.001
Male	16.1	22.0	28	-	26	67.0 *	33	-	0.761	
Female	16.8	22.3	28.9	-	20.4	85.7 *	14.3	-		
Vitamin A (mcg)										χ2(1) = 4.6978, *p* = 0.030
Male	602	894	1146	-	750	82.5 *	17.5	-	0.015	
Female	713	958	1263	-	750	88.5 *	11.5	-		
Vitamin C (mg)										χ2(1) = 18.5049, *p* < 0.001
Male	88	125	170	-	105	84.0 *	16	-	0.015	
Female	102	140	188	-	85	94.0 *	6	-		
Calcium (mg)										χ2(1) = 1.9735, *p* = 0.160
Male	476	690	961	-	1000	49.5 *	50.5	-	0.161	
Female	528	745	996	-	1000	55.3 *	44.7	-		
Iron (mg)										χ2(1) = 0.1467, *p* = 0.702
Male	12.0	16.4	21.7	-	8	99.0 *	.97	-	0.495	
Female	12.2	16.1	20.0	-	8	98.7 *	1.3	-		

RDA: Recommended Dietary Allowances. * Meeting 70% of RDA. ^ Sex differences in median nutrient intakes. # Sex differences in proportion of DRI consumed.

**Table 3 nutrients-15-03446-t003:** Frequency of positive answers to the questions constituting the NSI Determine Checklist, *n* (%).

NSI Determine Checklist Questions	All	Males	Females
1. I have an illness or condition that made me change the kind and/or amount of food I eat	164 (22.2)	44 (21.5)	120 (22.8)
2. I eat fewer than 2 meals per day	43 (5.8)	9 (4.4)	34 (6.4)
3. I eat few fruits or vegetable or milk products (less than once a day)	74 (10.0)	24 (11.8)	50 (9.4)
4. I have 3 or more drinks of beer, liquor or wine almost every day	6 (0.8)	4 (1.95)	2 (0.38)
5. I have tooth or mouth problems that make it hard to eat	37 (5.0)	10 (4.9)	27 (5.1)
6. I don’t always have enough money to buy the food I need	7 (0.90)	2 (0.97)	5 (0.94)
7. I eat alone most of the time	178 (24.1)	46 (22.3)	132 (24.8)
8. I take 3 or more different prescribed or over-the-counter drugs a day	225 (30.5)	68 (33.0)	157 (29.5)
9. I have lost 10 pounds (4.5 kg) in the last 6 months unintentionally	5 (0.70)	1 (0.49)	4 (0.75)
10. I am not always physically able to shop, cook and/or feed myself	2 (0.30)	0 (0.0)	2 (0.38)
Aggregate score in the NSI scale, mean ± SD	1.5 ± 2.1	1.5 ± 2.1	1.5 ± 2.1

**Table 4 nutrients-15-03446-t004:** Use of cooking oils, milk/milk substitutes and sugar/sweet options among older adults.

	n (%)	
Type of Oil Used for Cooking	Male	Female
Blended vegetable oil	96 (23.7)	239 (21.5)
Polyunsaturated oil	135 (33.3)	411 (37.0)
Monounsaturated oil	168 (41.5)	449 (40.4)
Saturated oil	6 (1.5)	12 (1.1)
*p*-value (Chi-square)	0.5	
Types of milk/milk substitute used in beverages	Male	Female
Creamer	9 (21.9)	181 (17.0)
Sweetened condensed milk	93 (21.5)	186 (18.0)
Evaporated milk	28 (6.5)	49 (4.7)
Full cream milk/power	14 (3.2)	52 (5.0)
Low-fat milk/powder	43 (9.9)	137 (13.2)
Skimmed milk/powder	3 (0.7)	23 (2.2)
No added milk	157 (36.3)	408 (39.3)
Whitener	0 (0.0)	1 (0.1)
Ask for less sugar/less sweet	Male	Female
No	231 (56.1)	565 (53.1)
Yes	181 (43.9)	499 (46.9)
*p*-value (Chi-square)	0.3	

**Table 5 nutrients-15-03446-t005:** Nutrition knowledge and acquisition.

		n (%)	
Scores to answering questions on ‘diet can be related to certain diseases.’	All	Male	Female
Low (0–3)	38 (5.1)	16 (7.8)	22 (4.1)
Moderate (4–6)	277 (37.5)	79 (38.4)	198 (37.2)
High (7 or above)	423 (57.3)	111 (53.9)	312 (58.7)
*p*-value (Chi-square)	0.1		
Do you pay attention to dietary and nutrition information?	All	Male	Female
Always	346 (47.8)	90 (44.6)	256 (49.0)
Sometimes	250 (34.5)	68 (33.7)	182 (34.9)
Seldom	102 (14.1)	35 (17.3)	67 (12.8)
I don’t know	26 (3.6)	9 (4.5)	17 (3.3)
*p*-value (Chi-square)	0.3		

Note: non-respondence treated as ‘I don’t know’.

**Table 6 nutrients-15-03446-t006:** Sources of nutrition information for older adults (multiple choice).

Rank	Sources of Nutrition Information	n (%)
1	book/newspaper/magazine, TV/radio, internet	571 (77.4)
2	relatives, friends and neighbors	236 (32.0)
3	medical practitioners, nurses, dietitians	122 (16.5)
4	courses or lectures	54 (7.3)
5	spouse	27 (3.7)
6	children	27 (3.7)
7	salespeople	17 (2.3)

## Data Availability

Requests for access to the DaHA data will be considered on a case-to-case basis. Request should be submitted to Lei Feng, who may be contacted by email (pcmfl@nus.edu.sg).

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
