# Peer review of "Adequacy of Nutrient Intake and Malnutrition Risk in Older Adults: Findings from the Diet and Healthy Aging Cohort Study"

_nutrients, 2023, doi:10.3390/nu15153446_

Round 1

Reviewer 1 Report

The Study provided a meaningful contribution to the field of research. The science underlying the main idea is strong. There are no significant gaps in the cited literature. The results are original and allow incremental advance over prior research results. The research design is appropriate. The right kinds of participants were used. The sample size was adequate. The correct statistics were used. The interpretation of the data makes sense and logically supports the conclusions. The findings are important and interesting to the readers. The methods are explained well enough that the experiments can be replicated. The discussion section intgrates the findings with relevant theory, rather than simply rehashing the introduction. The writing is good quality.

Overall, the study represents a promising starting point, but further refinement and expansion of the research could strengthen its impact and practical applicability in promoting healthy nutrition among older adults in the community.

Author Response

Dear Reviewer,

Thank you for your positive comments and time to review our manuscript. We will further use our findings to construct nutrition interventions for the local older adult population and conduct association studies to understand the role of nutrition in various health conditions. 

Best regards, 

Kaisy (on behalf of all authors)

*given no edit required from reviewer 1, there is no attachment

Reviewer 2 Report

I consider this article very relevant on the subject: “Diet quality and risk of malnutrition in older adults: findings from the Diet and Healthy Aging cohort study”

Which includes not only assessing the risk of malnutrition in older adults, but also assessing the intake of healthy food in a specific population.

I therefore make some suggestions that I believe may be a contribution to clarify and enrich the investigation:

I begin by emphasizing that diet quality should be replaced by healthy eating, as quality includes several dimensions that this article does not include.

Statistical analysis.

In my opinion the tables are unclear regarding statistical significance and correlations could have been made that could have enriched the article's discussion.

Material and methods

Paragraph 77,78 ……..The level of nutrient intake was evaluated according to the Recommended Dietary Allowances (RDAs). Note that the RDAs do not apply to people suffering from illnesses because it will influence nutrient intake. The RDAs only apply to healthy people.

Items 108 and 109 …….The questionnaire was prepared and validated by an internal research nutritionist…….how was it validated? Where is the reference? where is the questionnaire?

Statistical methods

Item 122……. Anthropometric data were analyzed……only BMI? Were weight and height not assessed?

Results

Paragraph 127……. or had dietary information missing or incorrect??.....

Paragraph 170 ……. (Table S2). It should be in the article and not attached.

Conclusions

What kind of proposal would you make for intervention in individuals detected at nutritional risk and what is the importance of this early identification?

Author Response

Response to Reviewer 2 Comments

Point 1: I begin by emphasizing that diet quality should be replaced by healthy eating, as quality includes several dimensions that this article does not include.

Response 1: Thank you for your comment. We agree that the term diet quality covers many aspects and we have not covered all. Therefore, we have decided to replace diet quality with ‘nutrient adequacy’ as adequacy of nutrient intakes is what we are measuring.

Point 2: Statistical analysis. In my opinion the tables are unclear regarding statistical significance and correlations could have been made that could have enriched the article's discussion.

Response 2: We have replaced figure 1 with a data table (table 2) to give clearer comparison between observed nutrient intakes and RDA, as well as statistical significance between sex. As the primary aim of our study was to just explore nutrient intakes, we did not do any correlation analysis. However, we are planning to do this for upcoming analysis.

Point 3: Material and methods Paragraph 77,78 ……..The level of nutrient intake was evaluated according to the Recommended Dietary Allowances (RDAs). Note that the RDAs do not apply to people suffering from illnesses because it will influence nutrient intake. The RDAs only apply to healthy people.

Response 3: Thank you for pointing that out. We have made the clarification accordingly. See line 88.

Point 4: Items 108 and 109 …….The questionnaire was prepared and validated by an internal research nutritionist…….how was it validated? Where is the reference? where is the questionnaire?

Response 4: We designed the questionnaire and were the first to utilize it, so the validity was not accessed. The questionnaire is now added to supplementary material for future references and validity testing.

Point 5: Statistical methods. Item 122……. Anthropometric data were analyzed……only BMI? Were weight and height not assessed?

Response 5: We did assessed weight and height. They are now included in Table 1.

Point 6: Results. Paragraph 127……. or had dietary information missing or incorrect??.....

Response 6: Yes, corrected

Point 7: Paragraph 170 ……. (Table S2). It should be in the article and not attached.

Response 7: Corrected

Point 8: Conclusions. What kind of proposal would you make for intervention in individuals detected at nutritional risk and what is the importance of this early identification?

Response 8: The use of NSI checklist for screening of malnutrition risk partly acted as an intervention for individuals at nutritional risk. The questions helped raise awareness in older adults of factors that affect their nutritional health, such as not eating adequate vegetables, or milk, have less than two meals a day and excessive alcohol drinking. Construction of public health interventions will be based on the risk factors identified through malnutrition risk screening, and this will allow more specific nutritional interventions for individuals at-risk. Please see changes made in text 306-310.
